# Advances in Adoptive Cell Therapies in Cancer: From Mechanistic Breakthroughs to Clinical Frontiers and Overcoming Barriers

**DOI:** 10.3390/medsci13030190

**Published:** 2025-09-15

**Authors:** Syed Arman Rabbani, Mohamed El-Tanani, Yahia El-Tanani, Rakesh Kumar, Shrestha Sharma, Mohammad Ahmed Khan, Suhel Parvez, Alaa A. A. Aljabali, Mohammad I. Matalka, Manfredi Rizzo

**Affiliations:** 1RAK College of Pharmacy, Ras Al Khaimah Medical and Health Sciences University, Ras Al Khaimah 11172, United Arab Emirates; 2Royal Cornwall Hospital, NHS Trust, Truro TR1 3LJ, UK; 3Amity Institute of Pharmacy, Amity University, Haryana 122412, India; 4Department of Pharmacy, Jagannath University, Haryana 124507, India; 5Department of Pharmacology, School of Pharmaceutical Education and Research, Jamia Hamdard, New Delhi 110062, India; 6Department of Toxicology, School of Chemical and Life Sciences, Jamia Hamdard, New Delhi 110062, India; 7Department of Pharmaceutics and Pharmaceutical Technology, Faculty of Pharmacy, Yarmouk University, Irbid 21163, Jordan; alaaj@yu.edu.jo; 8Department of Pathology and Microbiology, Faculty of Medicine, Jordan University of Science and Technology, Irbid 21163, Jordan; 9Department of Health Promotion Sciences Maternal and Infantile Care, Internal Medicine and Medical Specialties, School of Medicine, University of Palermo, 90127 Palermo, Italy

**Keywords:** T cell, CAR, immunotherapy, adoptive cell therapy, tumor infiltrating lymphocytes, multiple myeloma, solid tumors

## Abstract

Adoptive cell therapies (ACTs) have revolutionized cancer treatment by harnessing the specificity and potency of T lymphocytes. Chimeric antigen receptor (CAR)-T cells have achieved landmark successes in B-cell malignancies and multiple myeloma. Tumor-infiltrating lymphocytes (TILs) and T-cell receptor (TCR)-engineered T cells offer complementary strategies to target solid tumors and intracellular antigens. Despite these advances, ACTs face challenges including cytokine release syndrome, neurotoxicity, on-target/off-tumor effects, manufacturing scalability, and immunosuppressive tumor microenvironments. Innovative strategies, such as dual-antigen targeting, localized delivery, checkpoint blockade combinations, gene-editing, and machine-learning-guided antigen discovery, are being used to mitigate toxicity, enhance efficacy, and streamline production. As CAR-T, TIL, and TCR modalities converge with advances in manufacturing and computational biology, the next generation of “living drugs” promises broader applicability across hematologic and solid tumors, improved safety profiles, and better treatment outcomes for patients. This review details the evolution of ACTs from first-generation CAR constructs to next-generation “armored” designs. It also focuses on the development and clinical deployment of TIL and TCR therapies. Furthermore, it synthesizes mechanisms, pivotal clinical trial outcomes, and ongoing challenges of ACTs. It also highlights strategies that will drive broader, safer, and more durable applications of these therapies across hematologic and solid tumors.

## 1. Introduction

Effective immune surveillance is critical for preventing tumorigenesis; yet, many cancer patients exhibit impaired adaptive immunity that predisposes them to early malignant transformation [1,2,3]. Cancer immunotherapy seeks to strengthen host defenses by harnessing or engineering immune responses to selectively recognize and eliminate malignant cells. Early observations by William Coley on toxin-induced tumor regressions and the advent of monoclonal antibodies laid the groundwork for modern immuno-oncology [4]. Over the past decades, tumor biology, advances in immunology, and molecular engineering have led to a diverse arsenal of immunotherapeutic modalities, including adoptive T-cell transfer, immune checkpoint inhibitors, and therapeutic vaccines. These modalities offer unparalleled specificity, biocompatibility, and the potential for durable antitumor memory compared with conventional chemotherapy and radiotherapy [5].

The discovery in the 1960s that lymphocytes possess intrinsic cytotoxic capabilities against tumor cells catalyzed the study of T cells as therapeutic agents. In 1988, Rosenberg and colleagues achieved the first clinical proof of concept for an adoptive cell transfer by infusing autologous tumor-infiltrating lymphocytes (TILs) into melanoma patients [6]. Building on this, Zelig Eshhar introduced the first chimeric antigen receptor (CAR) in 1993, enabling T cells to recognize tumor-associated antigens independently of MHC restriction [6]. Following the addition of costimulatory domains CD28 in 1998 and 4-1BB/CD137 in 2004, CAR-T-cell proliferation, persistence, and antitumor effectiveness were significantly increased [7,8].

Adoptive cell transfer (ACT) encompasses the ex vivo isolation, expansion, and reinfusion of autologous or allogeneic immune cells endowed with antitumor specificity [9,10]. It has matured into three principal modalities: CAR-T cells, TILs, and TCR-engineered T cells. CAR-T cells use synthetic receptors attached to CD3ζ and one or more costimulatory domains (e.g., CD28 and 4-1BB) with a single-chain variable fragment for tumor antigen binding, enabling HLA-independent recognition of surface antigens and achieving profound remissions in hematologic malignancies. TILs are endogenous, polyclonal lymphocytes harvested from resected tumors, expanded in interleukin-2, and reinfused following lymphodepletion—capitalizing on the natural breadth of tumor antigen recognition and showing long-lasting responses in melanoma and other solid cancers. TCR-engineered T cells are autologous T cells genetically altered to express high-affinity T-cell receptors for intracellular tumor neoantigens presented by HLA molecules, extending targetable antigens to include differentiation, cancer-testis, and viral antigens [9,10].

Ex vivo expansion allows the selection of high-avidity effector cells and the incorporation of genetic or pharmacologic enhancements (e.g., suicide switches and cytokine armoring). Preconditioning regimens, such as lymphodepletion, further optimize engraftment. ACT offers precise dosing control, the potential for long-term persistence, and the flexibility to tailor cellular products compared with passive immunotherapies. Nevertheless, challenges such as antigen escape, on-target off-tumor toxicity, manufacturing complexity, and limited solid tumor infiltration remain significant [11,12].

Despite these hurdles, innovative strategies—including dual-antigen targeting, localized delivery, checkpoint blockade combinations, gene editing, and machine-learning-guided antigen discovery—are being applied to mitigate toxicity, enhance efficacy, and streamline production. As CAR-T, TIL, and TCR modalities converge with advances in manufacturing and computational biology, the next generation of “living drugs” promises broader applicability across hematologic and solid tumors, improved safety profiles, and better treatment outcomes for patients. This review details the evolution of ACTs from first-generation CAR constructs to next-generation “armored” designs and focuses on the development and clinical deployment of TIL and TCR therapies, synthesizing mechanisms, pivotal clinical trial outcomes, ongoing challenges, and strategies to drive broader, safer, and more durable applications of these therapies.

## 2. Fundamentals of CAR-T-Cell Therapy

CAR-T-cell therapy has made substantial advances in cancer treatment, notably for hematological malignancies. The U.S. FDA licensed tisagenlecleucel in 2017 for the treatment of pediatric and young adult patients with relapsed or refractory acute lymphocytic leukemia. Since then, six CAR-T-cell therapies have demonstrated remarkable efficacy in treating B-cell malignancies and multiple myeloma. Furthermore, the effectiveness of CAR-T-cell therapy against solid tumors remains limited. Notably, in February 2024, the FDA approved lifileucel, the first tumor-derived T-cell immunotherapy, for unresectable or metastatic melanoma, representing a significant milestone in addressing solid tumors. Despite these advancements, the high cost of CAR-T-cell therapies and their associated toxicities underscore the need for continued research to enhance safety profiles, reduce expenses, and expand applicability to a broader range of cancers [6,13].

CAR-T-cell therapy is recognized as one of the most advanced and explored approaches that plays a pivotal role in the development of cellular immunotherapy for the treatment of cancer [14]. In this technology, T lymphocytes are first isolated from the patient’s circulation. These cells are then genetically engineered with the desired genes to recognize and combat cancer cells without engaging MHC. The modified T cells are expanded ex vivo, after which they are infused back into the patient to elicit a targeted immune response against tumors. Post-infusion, patients are monitored for CAR-T-cell persistence and for disease responses, as illustrated in Figure 1 [15].

A key determinant of CAR-T efficacy lies in selecting antigens that are highly tumor-specific yet broadly expressed across malignant cells. While most current CARs target extracellular proteins such as CD19 or BCMA, insights from neoantigen research—tumor-specific peptides generated by somatic mutations—are shaping next-generation target discovery, particularly for solid tumors. Neoantigens presented on tumor cells via MHC molecules and recognized as “non-self” by T-cell receptors offer exceptional specificity with minimal off-tumor toxicity. Advances in next-generation sequencing, immunopeptidomics, and AI-driven epitope predictions now enable the identification of mutation-derived extracellular epitopes and membrane-associated targets amenable to CAR recognition. Integrating these neoantigen-informed strategies with combinatorial targeting and checkpoint blockade may help overcome antigen heterogeneity, immune escape, and the immunosuppressive tumor microenvironment—positioning them as a mechanistic bridge to expanding CAR-T success beyond hematologic cancers [11,14].

### 2.1. Generations of CAR-T-Cell Therapy

Mechanisms behind CAR-T-cell-mediated antitumor activity include perforin-granzyme-mediated cytolysis or the destruction of cancerous cells. Once the CAR-T cell becomes activated, it recognizes the target T cells on the surface antigen, which leads to the release of perforin and granzyme proteins. Further, perforin generates pores on the plasma membrane of T-cells, which allows the movement of granzymes that have cytotoxic potential into the T-cell cytoplasm, and this results in the destruction of antigen-positive malignant cells [15,16,17]. This revolutionary technology has transformed the management of hematological cancer and contributes to more than half of the cellular therapies commercially available and under development globally [18,19]. The construction of CARs has achieved significant development, from the first generation to the fifth generation (Figure 2), and the modifications in CAR constructions have been performed to improve efficacy and minimize associated toxicities.

#### 2.1.1. First Generation

This generation of cells comprises mainly four components: an extracellular domain for antigen binding, which is generally a single-chain variable fragment (scFv) made up of light or heavy chains of antibodies, a hinge (spacer) region, an intracellular signaling domain (CD3ζ), and a transmembrane. Structurally, first-generation CARs lack costimulatory molecules and cytokine-mediated signaling. These CARs are reported to have less T-cell proliferation, poor in vivo responses, and cytokine release, which result in reduced antitumor activity, hence making them obsolete [20,21].

#### 2.1.2. Second Generation

This generation of cells is engineered with one/two costimulatory molecule (CM) domains, such as CD28, CD134, CD19, or CD137 (4-1BB), along with an intracellular signaling domain (CD3ζ), to enhance the proliferation of T cells, the activation of T cells, survival, and cytotoxicity. All of the FDA-approved therapies that are commercially available are only from the second-generation CAR-T cells category, composed of a costimulatory molecule (CM) with a CD3ζ signaling domain [22]. As reported in various studies, CAR-T-cell therapy with the CD137 domain possesses a durable response, with constant persistence, but weak tonic signaling. In contrast, CAR-T cells with the CD28 domain have greater survival, T-cell expansion, memory cell formation, signaling, and quick responses. CAR-T cell therapies with CD19 domains are under clinical development and reported to be potentially effective against B-cell malignancies. Obecabtagene autoleucel, also known as “obe-cel,” is a novel CD19 CAR-T cell that was genetically engineered with CAT-41BB-Z. It is now undergoing clinical trials and is exhibiting exceptional outcomes in certain adult patients with recurrent B-cell acute lymphoblastic lymphoma (ALL) [23]. Regardless of these remarkable improvements, the challenges of persistence and relapse with CAR-T-cell therapies that have a single costimulatory domain continue, which has led to the development of third-generation cells [24].

#### 2.1.3. Third Generation

This generation of cells is constructed with a CD3ζ signaling domain combined with several CM domains, namely, CD27, CD28, CD134 (OX40), CD137, NKG2D, and TLR2, to create a CAR-T-cell integrated construct, such as CD3ζ-CD28-CD134, CD3ζ-CD28-CD137, and CD3ζ TLR2-CD28. Multiple additions of costimulatory molecules in the third generation help to overcome the constraint of a single costimulatory domain that is used in second-generation CARs; also, these additions provide the potent and faster elimination of tumors [25]. In some preclinical studies, it was found that third-generation CARs exhibited better antitumor activity and safety compared to second-generation CARs. However, the association of rapid CAR-T-cell exhaustion and severe side effects due to the overactivation of costimulatory molecules with third-generation CAR-T cells is common, which limits further application [26].

#### 2.1.4. Fourth Generation

This generation is also called T cells that are redirected for universal cytokine-mediated killing (TRUCKS), or armed CAR-T cells structurally based on the construct similar to the second generation, with the addition of transgenic proteins such as IL2, IL5, and IL12, and cytokine costimulatory molecules, such as CD27, CD28, CD134, and CD137 [27]. The presence of cytokines activates the innate immune cells that help to kill cancer cells and alleviate cancer. This type of CAR-T cell also helps in reestablishing the immune system of the cancer patient post-infusion. Even with these advantages, fourth-generation CARs have the limitation of moderate efficacy against solid tumors and are associated with some adverse events due to cytokine release in healthy tissues [28,29].

#### 2.1.5. Fifth Generation

Currently, the development of fifth-generation or next-generation CARs is in progress; structurally, they are like the construct of second-generation CARs, including a transcription-factor-binding motif and truncated cytoplasmic receptor [30]. This generation contains both advanced CARs, which outperform conventional CARs by adding extra structures to the standard CAR for the identification of numerous antigens or targets with low antigen density, and greatly enhanced conventional CARs, which are monovalent CARs that solely target one specific antigen [31]. The pros and cons of CAR-T cell generation are outlined in Table 1.

### 2.2. CAR-T-Cell Therapy for Different Malignancies

#### 2.2.1. Lymphoma

Hodgkin and non-Hodgkin are two major types of lymphoma that are traditionally treated with chemotherapy and monoclonal antibodies due to the limitation of disease deterioration that has been reported by patients in need of potential therapy. Both lymphomas have proven to be suitable to immunotherapy treatment because of their occurrence in immune-rich lymphoid tissues [32,33]. In various studies, CAR-T-cell therapy exhibited very promising results in B-cell lymphoma patients who were in a primary therapy setting [34]. In one study reported, CD19-antigen-directed CAR-T cells showed favorable outcomes in lymphoma, with patients achieving a 75% partial response. According to a clinical study by Kochenderfer et al., out of 15 patients with advanced B-cell malignancies who received CD19 CAR-T-cell therapy, 8 of them experienced complete remission, 4 showed partial remission, 2 were not responsive, and 1 had stable lymphoma [35]. In different studies, the treatment of lymphoma patients with CD19-targeted CAR-T cells has shown promising findings [36,37]. CD20-antigen-directed CAR-T cells have shown considerable results in the treatment of non-Hodgkin lymphoma patients [38,39]. Apart from the CD19 and CD20 antigens, CD30-targeted CAR-T cells have also been found to be vital in the treatment of lymphomas [40,41]. To understand and explore further the application of CAR-T cells, various clinical studies are in progress.

#### 2.2.2. Chronic Lymphocytic Leukemia

In recent times, various researchers have widely explored CAR-T-cell therapies in the treatment of patients with CLL, relapsed CLL, and relapsed or refractory (R/R) CLL. Among all CAR-T-cell therapies with CD19, antigens are extensively used. Some studies have demonstrated potential efficacy but limited proliferative responses and cell expansion abilities [42,43]. However, various toxic effects, such as tumor lysis and the depletion of B cells, are always associated with CAR-T-cell therapy. To overcome these various issues, researchers have explored the incorporation of interleukin and suicidal genes to improve antitumor efficacy and toxic effects, respectively [44,45].

#### 2.2.3. Acute Lymphoblastic Leukemia

Treatment with CAR-T-cell therapies has exhibited notable efficacy in patients with acute lymphoblastic leukemia (ALL), especially for curing fatal R/R B-ALL. Targeting the antigen CD19, which is a crucial marker of B cells, has become a productive approach, as the results have been satisfactory and the therapeutic effects have been in accordance with expectations. Kymirah was the first FDA-approved CD19-directed CAR-T-cell therapy used for the treatment of R/R ALL and B-cell lymphomas. However, due to the limitation of antigen escape, CD19-specific CAR-T cells are not responsive in some patients. To overcome this, researchers have explored CD20 and CD22 antigens for developing CAR-T cells and have found that both antigens can be a potential target in treating relapsed patients who are on CD19-directed CAR-T therapies [46,47,48].

#### 2.2.4. Multiple Myeloma

Multiple myeloma (MM) is a plasma cell malignancy often associated with anemia, hypercalcemia, immunosuppression, bone lesions, and renal failure [49]. The treatment of MM with conventional chemotherapy is challenging and is considered largely incurable due to molecular abnormalities, cellular heterogeneity, mutations in relapse malignant plasma cells, and high resistance to standard therapies. Recently, B-cell maturation antigen (BCMA)-targeted CAR-T-cell therapies, such as idecabtagene vicleucel and ciltacabtagene autoleucel, have emerged as promising options, demonstrating high response rates in relapsed or refractory MM. These advances highlight CAR-T-cell therapy as a novel and effective approach to overcoming therapeutic resistance in MM [50,51].

#### 2.2.5. Breast Cancer

CAR-T cells target 19 antigens for breast cancer, most of which are cell-surface proteins and members of the RTK family: HER2, AXL, EGFR, ROR1, HGFR/cMET, MUC1, CD70, MSLN, CD133, EpCAM, CD44v6, CSGP4, TEM8, ICAM1, TROP2, NKG2D, FRα, GD2, and CEA. While only 12 antigens have made it to clinical trials, all 19 have undergone considerable investigations and shown anticancer benefits in preclinical tests, such as tumor growth suppression and the release of pro-inflammatory cytokines. Although CAR-T-cell treatment for breast cancer has made significant progress thus far, individuals with the disease are still a long way from using it because of a lack of supporting data [52]. Key clinical outcomes of CAR-T cell therapies across different cancer types are illustrated in Figure 3.

#### 2.2.6. CAR-T-Cell Therapies in Solid Tumors

Recently, antigen-targeted CAR-T-cell therapies have displayed significant success in the treatment of various hematological malignancies, especially second-generation CARs, which have shown promising efficacy and an acceptable safety profile. CD19-targeted CAR-T cells are extensively explored for the indication of different lymphomas, myelomas, and B-cell leukemia [53,54,55]. Considering the success of CAR-T in hematological malignancies as a proof of concept, researchers have now focused on exploring the therapeutic potential of various surface proteins to treat solid tumors, namely, human epidermal growth factor receptors 1 and 2 (HER1 and 2), mesothelin (MSLN), fibroblast activation protein (FAP), diganlioside (GD2), L1 cell adhesion molecule (L1 CAM), mucin (MUC) 1, MUC 16, and vascular endothelial growth factor receptor 2 (VEGFR 2) [56,57]. Table 2 represents the clinical trials for anti-surface protein CAR-T-cell therapy against solid tumors.

### 2.3. CAR-T-Cell Therapies: Limitations and Potential Strategies

#### 2.3.1. Antigen Escape

It is defined as the partial or complete loss of the specific expression at targets by antigens, which results in tumor resistance. Antigen escape is considered one of the most challenging limitations associated with CAR-T-cell therapies and is generally observed in single-antigen-directed CAR structures. In various studies, it has been reported that patients with R/R ALL indications when treated with CD19-directed CAR-T-cell therapy showed durable response; however, the follow-up data revealed the downregulation of CD19 antigens, along with the development of disease resistance in 30–70% patients who had a recurrent disease post-therapy [58,59]. In multiple myeloma patients who have received BCMA-directed CAR-T-cell therapy, the loss of BCMA antigen expression was observed [60,61]. Table 3 outlines the key challenges encountered in CAR-T cell therapy and the strategies developed to address them.

To address this issue of antigen escape, several tactics have been used. Antigen-negative tumor variations are less likely to evade immune monitoring when CAR-T cells are engineered to identify many tumor-associated antigens at once. Approaches include pooled CAR-Ts (combining different CAR-T-cell populations), dual CAR-Ts (expressing two distinct CARs), and tandem CAR-Ts (single CAR constructs with dual antigen specificity). For instance, dual-targeting CAR-T cells against CD19 and CD22 has demonstrated enhanced efficacy in B-cell malignancies. The ability of dual-specific antibodies to bind to two distinct antigens makes it easier for immune effector cells to attach to malignant cells. Bispecific T-cell engager blinatumomab, which targets CD3 on T cells and CD19 on tumor cells, has shown promise in treating B-cell precursor acute lymphoblastic leukemia [63].

Integrating blinatumomab provides a parallel perspective on antigen-directed immunotherapies. While not genetically engineered like CAR-T cells, blinatumomab harnesses T-cell cytotoxicity through physical proximity between T cells and malignant cells, differing in manufacturing complexity, administration route, and in vivo persistence. This comparison situates CAR-T-cell therapy within a broader continuum of strategies aimed at overcoming antigen escapes and enhancing tumor-specific cytotoxicity [63].

#### 2.3.2. T-Cell Exhaustion

T-cell exhaustion, a dysfunctional state induced by chronic antigen stimulation, is a major factor limiting CAR-T-cell persistence and potency, particularly in solid tumors with sustained antigen exposure. It is marked by reduced proliferation, impaired cytokine production, and the continuous expression of inhibitory receptors such as PD-1, TIM-3, and LAG-3. These markers not only signify functional decline but also actively suppress CAR-T activity through inhibitory signaling pathways. To counter exhaustion, strategies include checkpoint blockades, the genetic disruption of inhibitory receptors, the optimization of CAR costimulatory domains to fine-tune activation, and metabolic reprogramming to maintain cellular fitness. Advanced approaches—such as transient CAR expression, combinatorial checkpoint targeting, and epigenetic reprogramming—are being explored to restore effector functions, prolong persistence, and expand efficacy in both hematologic and solid malignancies [44].

#### 2.3.3. Severe Adverse Effects

The treatment of hematological malignancies has seen significant progress due to CAR-T-cell therapy, but a significant concern with this strategy is the possibility of potentially fatal side effects. Among the most common adverse effects are cytokine release syndrome (CRS) and immune effector cell-associated neurotoxicity syndrome (ICANS). The cytokines IL-1 and IL-6, which mediate CRS, can result in fever, hypotension, and other systemic symptoms. For CAR-T-cell therapy, the FDA authorized the use of tocilizumab, a humanized anti-IL6 receptor antibody, to reduce the risk of CRS and neurotoxicity. The anti-IL6 chimeric antibody siltuximab and the IL-1 receptor antagonist anakinra are further possible therapies for these side effects [13]. Most cases of toxicities are reported in patients who were on second-generation CD19-directed CAR-T-cell therapy [70,71]. Besides these toxicities, renal failure, macrophage activation syndrome (MAS), and splenomegaly are other associated toxicities reported with CAR-T-cell therapies [72]. To counter the toxicity of CRS, IL-6 receptor blockade, along with corticosteroid and tocilizumab, which are approved by the FDA for the management of CRS, is used. Surprisingly, tocilizumab is associated with severe CRS and fatality [72,73,74]. To ameliorate toxicities associated with CAR-T-cell therapy, strategies such as altering the structure of CAR-T cells and engineering CAR-T cells are being explored.

#### 2.3.4. On Target Off-Tumor Effects

This type of toxicity occurs due to the direct attack on normal tissues; it happens because of the shared expression of the target antigens. This occurs in the case of solid tumor antigens when they are exposed or expressed on normal tissues at different levels. Hence, during CAR construction, the selection of specific antigen targets is crucial to ensure efficacy and avoid toxicities that arise due to on-target off-tumor effects. CAR-T-cell targets such as B7H3 [75,76], MUC1 [64], MUC16 [77,78], and TAG72 [79] are some major targets that are under investigation. A Phase I clinical trial (NCT05225363) of second-generation TAG72-targeted CAR-T cells is ongoing for the treatment of patients with platinum-resistant recurrent epithelial ovarian cancer [65]. Further developments and investigations with specific, targeted antigens are necessary to minimize toxicity and improve efficacy and therapeutic outcomes for CAR-T-cell therapies.

#### 2.3.5. Tumor Infiltration

According to research, immunosuppressive elements in the TME, including regulatory T cells, immunological checkpoint proteins, and suppressor cytokines, restrict the activation and growth of CAR-T cells. Furthermore, solid tumors typically have low vascular densities and high cell densities, which limit CAR-T-cell infiltration and spread and may lessen the tumor’s ability to fight it [80]. A study found that in a solid tumor mouse model, the combination of CAR-T-cell treatment and programmed cell death protein-1 (PD-1) inhibitors produced better therapeutic effects, effectively enhancing the CAR-T cells’ ability to survive and integrate. Furthermore, modifying CAR-T cells using gene-editing techniques to get beyond the TME’s limitations is another possible remedy. It has been discovered that even in the presence of large quantities of chemotherapeutic drugs, the cytotoxic capabilities of CAR-T cells can be improved by introducing genes that increase sensitivity towards antitumor drugs. Some researchers have employed the Siglec15 protein to test its therapeutic efficacy against lung cancer in nude mice. An anti-Siglec15 antibody was reported to improve the treatment of lung adenocarcinoma by increasing tumor cell death, presumably due to macrophage polarization altering the TME. cMet/Siglec15 CAR-T cells demonstrated clear cytotoxicity against lung cancer cells, both in vitro and in vivo. They could also dramatically reduce the development of lung adenocarcinoma cells in situ and enhance T-cell infiltration in TME.

To counteract these suppressive signals, combination therapies have been devised that integrate CAR-T-cell strategies with checkpoint inhibitors. For example, in B-cell lymphomas and hematological malignancies, the combination of CAR-T cells with anti-PD1 therapy is under clinical investigation (NCT05659628) [81]. Similarly, a Phase I/II study is exploring the safety and efficacy of MUC1 CAR-T cells combined with PD1-engineered T cells in patients with advanced esophageal cancer (NCT03706326) [82]. Despite the potential of these combination techniques, more investigations are required to completely evaluate their capacities to improve tumor infiltration and T-cell activity, especially when it comes to solid tumors and other complex malignancies [83,84,85]. Figure 4 represents the immunosuppressive features of the TME along with the mechanistic basis of combination immunotherapy.

#### 2.3.6. Overcoming Solid Tumor Barriers in CAR-T-Cell Therapy

Addressing the limitations of CAR-T-cell therapy in solid tumors requires a multipronged approach that integrates biological, engineering, and translational innovations. Armoring CAR-T cells with cytokine-secreting payloads such as IL-12 or IL-18 can reprogram the immunosuppressive tumor microenvironment, while the co-expression of dominant-negative receptors or checkpoint inhibitors (e.g., PD-1 decoys) mitigates T-cell exhaustion. Engineering dual- or multi-target CARs enhances antigen coverage and reduces the risk of tumor antigen escape. Modifying chemokine receptors to match tumor-secreted chemokines (e.g., CCR2 for CCL2-rich tumors) improves homing and infiltration. Incorporating logic-gated CAR designs and switchable safety modules enhances precision and minimizes off-tumor toxicity. Beyond cell-intrinsic engineering, combining CAR-T therapy with oncolytic viruses, radiotherapy, or immune checkpoint blockades may synergistically improve efficacy against solid tumors. Together, these strategies aim to transform the hostile architecture of solid tumors into an accessible and immunologically favorable battlefield for CAR-T cells [57,85].

#### 2.3.7. Integration of Artificial Intelligence and Machine Learning in CAR-T Cell Development

Recent advances in artificial intelligence (AI) and machine learning (ML) are increasingly being leveraged to optimize CAR-T-cell therapy design and clinical application. AI-driven algorithms can analyze high-dimensional omics datasets to identify optimal tumor-associated antigens with minimal off-target risks, enabling more precise target selection. Machine-learning models are also being used to predict CAR–antigen-binding affinities, optimize single-chain variable fragment (scFv) sequences, and model intracellular signaling domains for enhanced therapeutic efficacy and persistence. Furthermore, AI-enabled image analysis and real-time patient monitoring tools can assist in early detection and management of cytokine release syndrome (CRS) and neurotoxicity, improving patient safety. By integrating AI/ML across target discovery, construct engineering, and clinical monitoring, CAR-T-cell therapies can be made more effective, personalized, and adaptable to tumor evolution—potentially overcoming key limitations such as antigen escape and heterogeneous tumor expression [82,83].

#### 2.3.8. CAR-T-Cell Therapy—Economic and Logistical Challenges

CAR-T-cell therapy is constrained by prohibitively high costs—often driven by individualized autologous manufacturing, GMP-grade facilities, and complex cold-chain logistics. Limited availability of specialized centers, intensive monitoring requirements for toxicity management, and long vein-to-vein times further restrict access, especially in low-resource settings. Emerging solutions such as decentralized point-of-care manufacturing, automation, and off-the-shelf allogeneic CAR-T products aim to improve affordability and scalability [80,81].

## 3. Fundamentals of Tumor Infiltration Lymphocyte Therapy (TIL)

Metastatic melanoma patients have long been treated with ACT using autologous TILs. Since TILs are extracted from the patient’s own tumor tissue and reinfused following in vitro activation and proliferation, TIL therapy represents the most individualized treatment and capitalizes on naturally occurring tumor-reactive T cells that are already present within the tumor. Interleukin-2 (IL-2) infusions come after preconditioning lymphodepleting chemotherapy, which comes before TIL infusions. Several independent institutes have reported objective response rates of up to 50%, with 10–20% of patients experiencing complete tumor shrinkage [85].

TILs, a heterogeneous population of polyclonal cells with a high capacity for tumor cell identification, are lymphocytes that have entered the tumor via the bloodstream. Immune cells are isolated and grown outside of the tumor microenvironment in vitro as part of TIL therapy (Figure 5). TIL therapy involves the collection of T cells from the patient’s tumor, whereas CAR-T-cell production involves the collection of T cells from the patient’s blood circulation. TILs are extracted from removed tumors and then chosen, separated, and grown as part of the procedure. TILs have the ability to combat cancer because they are not genetically altered; instead, they are grown in vitro into enormous numbers, ranging from hundreds to billions of cells. When necessary, the cryopreserved TIL product is reinfused into patients along with chemotherapy that depletes their lymph nodes.

### 3.1. TIL Therapy for Cancer

#### 3.1.1. Solid Tumors

In 1988, the first in-human TIL therapy was performed for the treatment of patients with metastatic melanoma; lymphocytes were extracted from resected melanoma, and after in vitro expansion, they were infused back into the melanoma patients, followed by interleukin 2 (IL-2) as an adjuvant therapy and cyclophosphamide as chemotherapy [86,87]. The results obtained were promising, with an OR rate of 55% and a median overall survival (OS) of approximately 11 months. The findings of this study demonstrated the potency of the TIL therapy in the elimination of tumor cells, along with an acceptable safety profile [88,89,90]. As of now, many researchers have explored the application of TIL therapies for the treatment of advanced melanoma (NCT06703398, NCT06120712, NCT05098184) [91], advance malignant solid tumors, pediatric solid tumors (NCT06047977) [92], advanced gynecological tumors (NCT05098171, NCT05468307, NCT04766320), R/R Gynecologic tumors (NCT04766320) [93], and non-small-cell lung cancer (NCT06473961, NCT06455917) [94].

#### 3.1.2. Non-Small-Cell Lung Cancer (NSCLC)

NSCLC has shown the most potential for TIL therapy among cancer types. In a Phase I pilot study at Moffett Cancer Center, 20 patients with NSCLC had their tumors harvested for TIL generation before being exposed to PD-1. At least two months of nivolumab (four cycles of 240 mg every two weeks) was then administered to them while their TILs were being produced. Preliminary outcomes showed an ORR of 25%, a disease control rate (DCR) of 60%, and a median progression-free survival (PFS) of 5.8 months. This was the first TIL encounter in NSCLC that was recorded, suggesting that TIL treatment may result in notable and durable outcomes for NSCLC [95].

#### 3.1.3. Cervical Cancer

The NCI’s Phase II study of TIL therapy for cervical cancers included a group of 18 patients whose lesions tested positive for HPV 16 or HPV 18. The trial reported an ORR of 28%, including complete responses (CR) in two patients and partial responses (PR) in three, with a median duration of response (DoR) exceeding 15 months. TIL treatment was examined in C-145-04, a different Phase II multicenter clinical trial from Iovance, in patients with advanced cervical cancer that was ICI-naïve and had advanced following at least one previous line of chemotherapy. This study showed an ORR of 44%, with CR in 11%, and a median OS of approximately 12 months. In order to accelerate the development of TIL therapy, the FDA granted breakthrough therapy designation status based on these findings [96]. The stepwise process of tumor-infiltrating lymphocyte (TIL) therapy production is illustrated in Figure 5.

#### 3.1.4. Breast Cancer

TIL treatment is unlikely to target breast cancer. A total of 42 breast cancer patients who had not responded to previous treatments were involved in Phase II research undertaken by the NCI; the clinical benefit was limited, with an ORR below 5%, and a median progression-free survival (PFS) of 3 months. All patients’ tumor harvests were used to create TIL cultures, which were subsequently further chosen based on their reactivity to distinct neoantigens found in the patients’ tumors. Enhancing TIL for neoantigen-reactive populations to boost potency is of great interest [97].

There is a significant comparison between CAR-T-cell therapy and TIL therapy. TIL therapy is, basically, more favorable for the treatment of solid tumors that have more mutational burdens, like melanoma or certain types of lung cancer, where native immune responses exist but require amplification [98]. On the other hand, CAR-T-cell therapy is more effective and widely used in hematological malignancies with known and consistent antigen targets. The decision to use one approach over the other depends on multiple factors, including the tumor type, antigen expression, patient-specific immune profiles, and resource availability [99]. While CAR-T-cell therapy continues to dominate the hematologic cancer space, advances in TIL therapy are helping extend cellular immunotherapy to a broader range of solid tumors. Table 4 outlines the key ongoing clinical trials investigating CAR-T cell and T-cell therapies.

## 4. TCR-Engineered T Cells

T-cell immunotherapy primarily uses T cells that have been engineered to bind to the CAR and the T-cell receptor (TCR). CAR-T-cell therapy has produced promising results in clinical trials, and numerous medicines have been approved for the treatment of hematological malignant tumors. Its known effects on solid tumors are still inadequate, but for the treatment of solid malignancies, TCR-T-cell therapy has shown promising results thus far. Comprising two highly variable heterogeneous peptide chains connected by disulfide bonds, TCR is a protein found on the surface of T cells that specifically detects and mediates immune responses. The four peptide chains that make up TCRs are α, β, γ, and δ. αβ TCRs are formed by α and β peptide chains, whereas γδ TCRs are formed by γ and δ peptide chains. αβ TCRs start the TCR signaling cascade by binding to MHC on tumor cells or antigen-presenting cells (APCs). Following this, a number of intracellular proteins that mediate T-cell immunological activity are activated, including 70-kD zeta-associated protein (ZAP70), nuclear factor of activated T cells 2 (NFAT2), and CD3ζ [100].

T cells are genetically engineered with a TCR gene sequence that specifically detects tumor antigens, resulting in TCR-T cells that can precisely cause cytotoxicity to tumor cells. TCR-T cells are able to identify a greater range of target antigens because they are able to identify both intracellular and specific antigens on the surface of tumor cells [101,102]. Multiple myeloma, metastatic melanoma, and metastatic synovial sarcoma can all be treated using TCR-T-cell therapy that targets NY-ESO-1 specifically. It has shown encouraging outcomes, antigen-specific and multifunctional action, and long-lasting antitumor responses [103]. The application of targeted TCR-engineered cells across different cancer types is summarized in Table 5.

### 4.1. TCR-Targeted Tumor Antigens

#### 4.1.1. Tumor Associated Antigens (TAA)

Although tumor cells express TAAs, at least some healthy tissues also do. Treatments that target TAAs must therefore deal with the possibility of T-cell-mediated on-target off-tumor toxicity. Both normal cells of the same tissue origin and malignant cells display differentiation antigens. Melanoma-differentiation antigens, which include the extensively researched melanoma-associated antigen recognized by T lymphocytes (MART-1) [113] and glycoprotein 100 (gp100) [114], were among the first tumor antigens to be identified. Cancer cells express overexpressed antigens in high quantities, whereas healthy cells express them at low levels. Wilms’ Tumor Antigen 1 (WT1), a transcription factor that is produced 10–1000 times more frequently in leukemic cells than in normal cells, is an example of an extensively researched overexpressed antigen [115,116].

#### 4.1.2. Cancer-Germline Antigens (CGAs)

Since CGAs are only expressed in germline cells, such as testis cells, which do not express HLA-class I, they significantly lower the chance of on-target off-tumor damage. In contrast, CGAs are aberrantly expressed in cancer cells. NY-ESO-1 and MAGE-A4, which are found to be expressed at high levels in a variety of solid and hematological malignancies, are examples of CGAs with high clinical significance [117].

#### 4.1.3. Tumor-Specific Antigens (TSAs)

TSAs are not found in the genomes of any normal cells, but they are genetically encoded in cancer cells. TSAs can also be categorized as neoantigens or viral antigens. Since the expression of viral oncogenes is almost nonexistent in normal cells and frequently homogenously expressed in virus-driven malignancies, they make very appealing targets for tumor antigens. Clinically significant viral antigens include EBV viral oncogenes LMP1 and LMP2, which are expressed in several solid and hematological malignancies [117,118], and the HPV viral oncogenes E6 and E7, which are expressed in several forms of epithelial carcinoma [106].

### 4.2. Challenges of TCR T-Cell Therapies

#### 4.2.1. Toxicity Prediction

Immunopeptidomics, the bioinformatics analysis of transcriptomic and proteomic databases, and in vitro or ex vivo tests are used to assess TCR-T-cell-related on-target off-tumor toxicity and ascertain whether TCR-T cells are able to recognize normal cells or tissues. TCR recognition of an antigen other than the target antigen on normal cells is associated with off-target off-tumor toxicities or cross-reactivity. Clinical experiments employing affinity-enhanced TCR-T cells have documented cross-reactivities [119].

#### 4.2.2. Resistance

The main concern is whether TCR-T cells acquire secondary resistance mechanisms. By activating immune checkpoint receptors, immune checkpoint ligands on tumor cell surfaces may be overexpressed, which could impair transplanted T-cell proliferation and function and ultimately cause fatigue. The main mechanism that causes tumor cells to evade TCR-T-cell therapy is by losing or reducing MHC class I molecules, which prevent TCR-T cells from identifying the target epitope. A relapse occurred six months following TCR-T-cell therapy in a clinical study that targeted P53 mutations in a patient with breast cancer. The tumor cells displayed LOH of chromosome 6, with the HLA-A*02:01 locus and expressed intact P53 [119].

#### 4.2.3. Tumor Microenvironment

The three primary components of the tumor microenvironment are immunosuppression, persistent inflammation, and hypoxia. Although CD8+ T lymphocytes are essential for the removal of tumors, they become weaker and eventually fatigued when exposed to hypoxic environments. Recent findings have revealed a paradox: as this essential component runs out, it changes, increasing the expression of CD39 to create an immunosuppressive environment that weakens other T cells’ potent anticancer properties. Programmed death ligand-1 (PD-L1) is one of many immunosuppressive signaling proteins present in the majority of tumor stromal cells in the TME. Immune cell malfunction and apoptosis may arise from the interaction of PD-L1 with T-cell-expressed programmed cell death protein-1 (PD-1). In necrotic tumor microenvironment regions, hypoxia causes an acidic environment and increased potassium levels, which upset potassium homeostasis in tumor-infiltrating lymphocytes (TILs). This significantly reduces T-cell effector function, which affects the ability and activity of cytokine release [103].

#### 4.2.4. TCR-Engineered T-Cell Therapy—Economic and Logistical Challenges

TCR-engineered T-cell therapy faces similar barriers, with high production costs from patient-specific cell isolation, genetic modification, and expansion under GMP conditions. Additional complexity arises from mandatory HLA typing and antigen validation, limiting timely access in aggressive disease. In low- and middle-income regions, there is inadequate infrastructure, and trained personnel exacerbate inequities, highlighting the need for streamlined manufacturing, shared international production hubs, and cost-efficient vector generation strategies [101,102].

### 4.3. Integration of Artificial Intelligence and Machine Learning in TCR-Engineered T Cells

The development of TCR-engineered T cells critically depends on identifying high-affinity, tumor-specific TCR–peptide–MHC interactions while avoiding cross-reactivity with healthy tissues. Artificial intelligence (AI) and machine learning (ML) are increasingly transforming this process by enabling the rapid, accurate, and large-scale prediction of immunologically relevant epitopes [99].

#### 4.3.1. Epitope Prediction and Prioritization

Deep-learning platforms such as *NeoaPred* have achieved >80% accuracy in predicting peptide–HLA binding and immunogenicity, streamlining the identification of viable TCR targets from vast genomic datasets.

#### 4.3.2. TCR–pMHC Interaction Modeling

ML-based structural prediction tools are enhancing in silico screening of TCRs against predicted epitopes, enabling the early detection of potential off-target cross-reactivity and improving therapeutic safety.

#### 4.3.3. Personalized Target Discovery

By integrating patient-specific tumor sequencing data with AI algorithms, novel neoantigen targets can be identified and prioritized, particularly for rare HLA alleles, where traditional databases are sparse [99].

These computational advancements shorten the discovery pipeline, improve the precision of TCR design, and lay the foundation for next-generation TCR therapies that are more personalized, effective, and safe.

## 5. FDA-Approved ACTs

Currently, six CAR-T cells and two TILs have been approved by the FDA for the treatment of R/R ALL, large B-cell lymphomas, R/R multiple myeloma, R/R B-cell precursor acute lymphoblastic leukemia (AML), and unresectable or metastatic melanoma.

### 5.1. Tisagenlecleucel

Tisagenlecleucel, also known as CTL019, tisacel, or Kymirah, is the first FDA-approved CD19-directed second-generation CAR-T-cell therapy. It was approved on 30 August 2017, and Novartis brought it to market for the treatment of R/R acute lymphoblastic leukemia in children and young adults between the ages of 3 and 25 years. Later, in May 2018, it was approved for the treatment of adult patients with B-cell lymphomas. Kymirah is also used for the treatment of R/R large B-cell lymphomas, high-grade B-cell lymphomas (HGBCL), diffuse large B-cell lymphomas (DLBCL), and in adult patients with R/R follicular lymphomas post-systemic therapy (two or more lines). Kymirah has limitations when it comes to treating individuals with primary central nervous system lymphomas and chronic lymphocytic leukemia, despite its significant efficacy and safety [120,121].

### 5.2. Axicabtagene Ciloleucel

Axicabtagene ciloleucel, also known as KTE-C19, Axi-Cel, or Yescarta, is the second approved CD19-targeted second-generation CAR-T-cell therapy used for adult patients with large B-cell lymphomas (primary mediastinal large B-cell lymphomas, HGBCL, including DLBCL, not otherwise specified); it received approval by the FDA on 18 October 2017 and was made commercially available by Kite Pharma [122]. As reported in real-world data, patients with HGBCL on Yescarta therapy have shown 64% CR and 82% OR [123]. In another study, Yescarta was found to be highly effective, with 78% CR and an 89% OR rate when administered as a first-line therapy for the indication of high-risk DLBCL [124].

### 5.3. Brexucabtagene Autoleucel

Brexucabtagene autoleucel, KTE-X19, or Tecartus is the third approved CD19-directed CAR-T-cell therapy used for the indication of leukemia and lymphoma. Tecartus received its FDA approval on 24 July 2020 for the indication of R/R MCL and was made commercially available by Kite Pharma. It was also approved for R/R B-cell ALL in adult patients on 1 October 2021 [125]. In different clinical studies, Tecartus has shown promising responses, along with significant outcomes, with a complete remission of 67% and an OR rate of 93% post-12-month follow-up in patients, with 61% and 83% progression-free survival (PFS) and overall survival (OS), respectively. However, Tecartus therapy is also associated with some major adverse events, such as cytopenia, neurotoxicity syndrome, infection, and cytokine release syndrome, which limit its use [126,127].

### 5.4. Lisocabtagene Maraleucel

Lisocabtagene maraleucel is available commercially as liso-cel or Breyanzi and is a CD19-directed CAR-T-cell therapy that received its approval by the FDA on 5 January 2021 for the treatment of large B-cell lymphoma in adult patients post-systemic therapy (two or more lines), HGBCL, DLBCL, primary mediastinal large B-cell lymphoma, and grade 3b follicular lymphoma patients [128]. Two-thirds of patients with B-cell lymphomas on Breyanzi therapy have demonstrated remission, lasting between 6 and 9 months. Contrary to this, it is also associated with adverse events such as cytopenia, neurotoxicity, and cytokine release syndrome [129,130].

### 5.5. Idecabtagene Vicleucel

Idecabtagene vicleucel or bb2121 is available commercially as Abecma and is a B-cell maturation antigen (BCMA)-directed CAR-T-cell therapy, first approved by the FDA on 26 March 2021 for the treatment of R/R MM (post four or more prior systemic therapies). Abecma therapy in R/R MM patients has demonstrated significant improvement, with a 73% OR rate, 33% CR, and significant survival, with 8.8, 10.7, and 19.4 months of median response durations for PFS and OS, respectively. The interesting thing about Abecma is that it accounts for very few side effects compared to other approved CAR-T-cell therapies [131,132,133].

### 5.6. Ciltacabtagene Autoleucel

Ciltacabtagene autoleucel is available commercially as Carvykti or Cilta-cel and is a BCMA antigen-targeted CAR-T-cell therapy; it received its approval by the FDA on 28 February 2022 for the treatment of R/R MM patients post prior lines of therapy (four or more) [134]. The safety and efficacy of Carvykti is known in regard to the findings of the CARTITUDE-1 trial (Phase 1b/2), where patients on the therapy with R/R multiple myeloma demonstrated a 98% OR rate, 80% CR rate, and 18 and 21.8 months of PFS and median response duration, respectively [135,136].

### 5.7. Obecabtagene Autoleucel

Obecabtagene autoleucel is commercially available as Aucatzyl or obe-cel and is a CD19-directed CAR-T-cell therapy recently approved by the FDA on 8 November 2024 to use for the indication of R/R B-cell precursor AML in adult patients [137]. Aucatzyl provides a rapid binding to the CD19 domain, which results in a shorter period of CAR-T-cell binding to leukemia cells to provide a reduced rate of T-cell exhaustion and cytokine secretion. The safety and efficacy of Aucatzyl were based on the findings of the Phase 1/2 FELIX trial, out of 65 patients with R/R AML who had received the therapy, 27 patients showed CR within 3 months, and the median duration of CR was 14.1 months. Cytokine release syndrome, ICANS, and T-cell malignancies are the most frequently documented side effects of Aucatzyl therapy, occurring in 75%, 64%, and 24% of patients, respectively [138].

### 5.8. Lifileucel

Lifileucel, Amtagvi, or LN-144 is the first approved cryopreserved autologous TIL therapy. It received approval by the FDA on 16 February 2024 for the treatment of unresectable or metastatic melanoma in adult patients who were on a post-PD-1 blocking antibody and BRAF inhibitor (with/without a MEK inhibitor) setting [139]. Approval of Amtagvi was based on the safety outcomes and efficacy findings of a clinical trial (NCT02360579) that was sponsored by Iovance Biotherapeutics. The findings showed that among 73 patients who received the therapy, they achieved an OR rate of 31.5%, and the initial duration of response was 1.5 months [140]. A comparative analysis of different ACT modalities is presented in Table 6.

## 6. Future Perspectives

Advances in the construct of engineered next-generation CAR-T cells will pave the way towards unexplored innovative signaling pathways that will significantly broaden the therapeutic potential of CAR-T cells. Currently, CAR-T cell therapy has proven its efficacy and safety in patients with hematological malignancies. Continuous research will provide a better understanding of molecular and mechanistic developments, which will ultimately play a pivotal role in broadening the functionality and application of CAR-T cells in the next decade. However, the extensive therapeutic application of CAR-T cells in solid tumors is still restricted due to the challenges of cell trafficking, immunosuppressive tumor microenvironment, and infiltration at the tumor site.

These challenges can be dealt with by utilizing the local administration of the therapy at the tumor, as discussed in the review. The combination of utilizing various antigens, along with co-inhibitory pathways, holds potential capability for the construction of more advanced CAR-T-cell therapies. In spite of these advances, there is still a need to explore surface-protein-targeted CAR-T-cell therapy. As shown in some early clinical studies, MSLN-, HER2-, GD2-, and FAP-targeted CAR-T cells have demonstrated promising therapeutic potential for the treatment of various solid tumors and complex malignancies by minimizing toxicities and improving efficacy among cancer patients.

The bulk of clinical responses to TIL therapy in patients with melanoma are most likely caused by neoantigen-specific T-cell reactivity, according to data gathered in recent years. Accordingly, TIL therapies have been primarily preferred over alternative T-cell therapy approaches that employ CARs or TCRs since the former target several neoantigens at once, potentially lowering the risk of tumor escape [85].

TCR-T-cell therapies are moving away from the tailored design of particular antigens and toward the allogeneic generality of this idea, which is a step forward in scientific studies. To address the issue of insufficient targets for TCR-T-cell treatment, membrane-fused nanoparticles (NPs) have been suggested as targets for the specific identification of TCR-T cells, regardless of the original HLA type. This will alter the peptide-HLA (pHLA) on tumor cell surfaces, making it possible to identify and remove tumor cells specifically [103].

Artificial intelligence (AI) and machine learning (ML) are emerging as transformative tools in adoptive cell therapy (ACT) research, enabling data-driven precision across the entire development pipeline. By integrating multi-omics datasets—genomics, transcriptomics, proteomics, and immunopeptidomics—AI/ML algorithms can predict optimal CAR/TCR designs, identify patient-specific neoantigens, and forecast antigen escape variants before they arise. Neural networks facilitate the construction of CAR constructs with optimized signaling motifs, accelerating development timelines. In manufacturing, AI-driven image analysis and process control systems enhance quality assurance by continuously monitoring the cell phenotype, viability, and functional potency. Clinically, predictive models based on pre-infusion transcriptomes can anticipate patient responses, as well as risks of cytokine release syndrome, neurotoxicity, and therapy resistance, enabling proactive interventions. Furthermore, natural language processing rapidly synthesizes insights from the expanding immuno-oncology literature, guiding rational trial design. As these computational frameworks mature, their integration into experimental and clinical pipelines promises to improve safety, personalization, scalability, and accessibility of ACTs [141,142].

Future ACT development should integrate patient-reported outcomes and quality-of-life metrics alongside traditional survival endpoints. While current evidence suggests that durable responders to CAR-T-cell and TCR-engineered T-cell therapies often experience improved functional capacity, reduced treatment burden, and enhanced psychological well-being, transient toxicities such as CRS and neurotoxicity can temporarily affect daily living. Incorporating Quality-of-Life (QoL)-focused trial designs and supportive care strategies will be essential to ensure that therapeutic advances translate into meaningful, sustainable improvements in patients’ overall well-being.

## 7. Conclusions

Adoptive cell therapies have established a new therapeutic paradigm in oncology, achieving unprecedented remissions in hematologic malignancies and offering the first durable responses in select solid tumors. Through the iterative engineering of antigen receptors, costimulatory domains, and transcriptional circuits, CAR-T platforms have delivered life-saving benefits in B-cell malignancies and multiple myeloma. Parallel advances in TIL and TCR therapies have broadened the target landscape to include naturally tumor-reactive repertoires and intracellular neoantigens, respectively. This is extending ACT applicability across melanoma, sarcoma, and virus-associated cancers. Despite these milestones, ACTs still face hurdles such as cytokine release syndrome, neurotoxicity, antigen heterogeneity, immunosuppressive microenvironments, and the logistical complexity of personalized manufacturing. Emerging strategies such as multi-antigen targeting, “armored” constructs, localized delivery, checkpoint-inhibitor combinations, and CRISPR-mediated “stealth” phenotypes are actively addressing these barriers.

AI now stands poised to accelerate every phase of ACT development. ML algorithms can mine tumor and HLA-peptidome datasets to predict high-affinity neoepitopes and optimize CAR/TCR-binding domains. Predictive models trained on clinical and single-cell transcriptomic data are improving patient selection and forecasting toxicity risks. In manufacturing, AI-powered image analysis and process-control systems are streamlining cell-expansion workflows, enhancing lot-to-lot consistency, and reducing time to release. Digital-twin simulations of cell–tumor interactions promise the ability to forecast in vivo persistence and efficacy.

Looking forward, the synergistic integration of synthetic biology, precision gene editing, computational immunology, and AI-augmented workflows will drive ACTs toward broader accessibility, enhanced safety, and better treatment outcomes. Continued collaboration among clinicians, engineers, and data scientists will be essential to fully realize ACT’s transformative potential in cancer therapy.

## Figures and Tables

**Figure 1 medsci-13-00190-f001:**
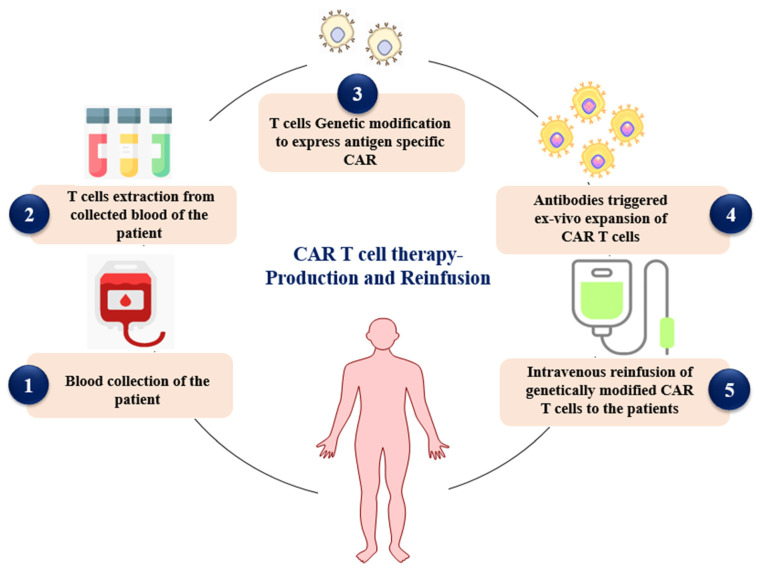
Diagrammatic representation of the process CAR-T production and reinfusion to the patients (Created in BioRender. Rabbani, S. (2025) https://BioRender.com/y6vtd5g (accessed on 24 July 2025)).

**Figure 2 medsci-13-00190-f002:**
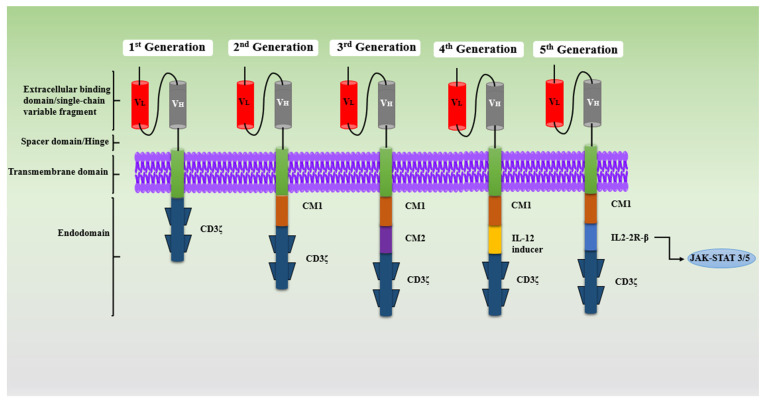
Diagrammatic presentation of modification in the chimeric antigen receptor (CAR) construct from first generation to fifth generation (Created in BioRender. Rabbani, S. (2025) https://BioRender.com/pcqbpeu (accessed on 24 July 2025)).

**Figure 3 medsci-13-00190-f003:**
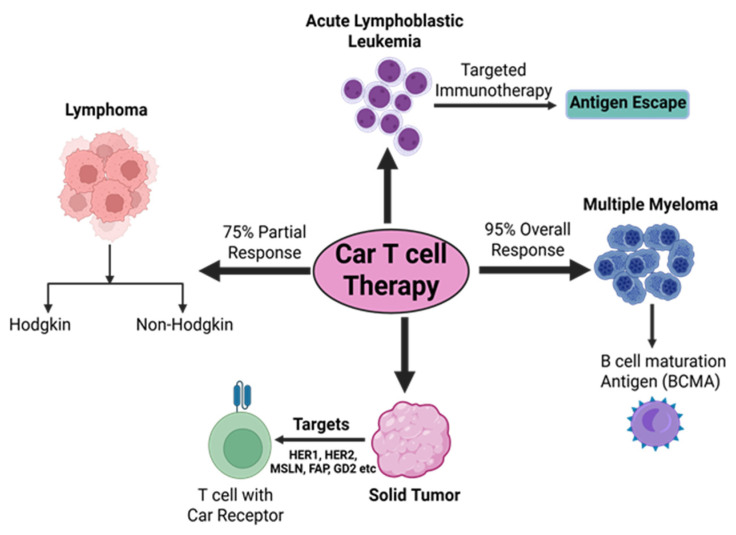
Figure depicting key clinical outcomes of CAR-T-cell therapies among various cancer types (Created in BioRender. Rabbani, S. (2025) https://BioRender.com/bj3yoxd accessed on 24 July 2025).

**Figure 4 medsci-13-00190-f004:**
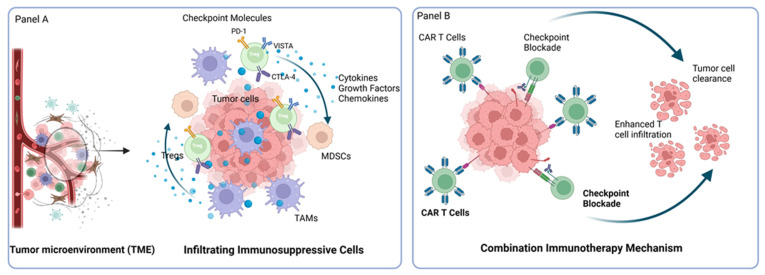
Representation of the Immunosuppressive TME and the Mechanism of Combination Immunotherapy (Created in BioRender. Rabbani, S. (2025) https://BioRender.com/hqoivh4 accessed on 24 July 2025). Panel (**A**) illustrates the immunosuppressive tumor microenvironment where tumor cells recruit and are surrounded by key immunosuppressive cell types—regulatory T cells (Tregs), tumor-associated macrophages (TAMs), and myeloid-derived suppressor cells (MDSCs). These cells secrete a variety of growth factors, cytokines, and chemokines, and contribute to the upregulation of immune checkpoint molecules such as PD-1, VISTA, and CTLA-4, collectively dampening antitumor immune responses. Panel (**B**) depicts the therapeutic strategy where CAR-T cells are combined with checkpoint inhibitors (e.g., anti-PD1). This combination is designed to block the inhibitory signals, enhance T-cell infiltration, and reactivate cytotoxic activity against tumor cells, thereby overcoming the immunosuppressive barriers within the tumor microenvironment.

**Figure 5 medsci-13-00190-f005:**
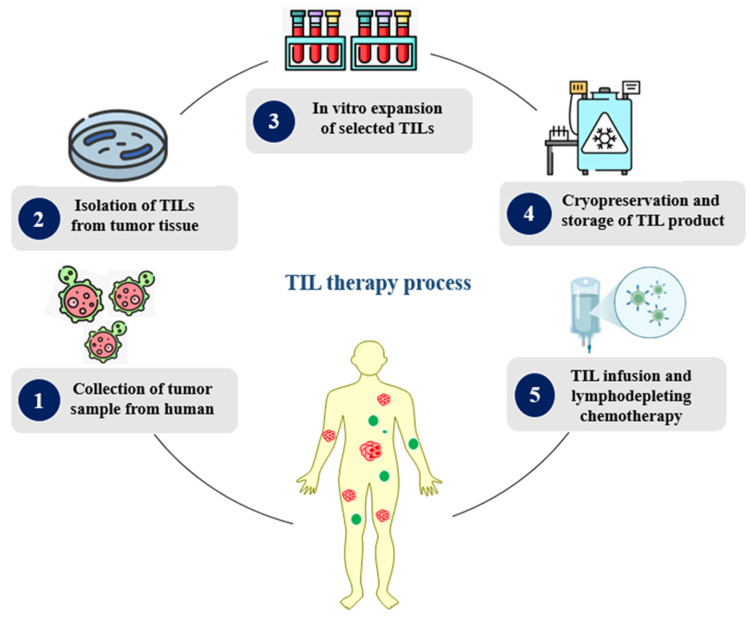
Diagrammatic representation of the process of TIL therapy production (Created in BioRender. Rabbani, S. (2025) https://BioRender.com/632n5f1 accessed on 24 July 2025).

**Table 1 medsci-13-00190-t001:** Comparative data of the pros and cons related to development of CAR-T-cell generation.

CAR Generation	Components	Key Features	Pros	Cons	References
First	scFv, spacer, CD3ζ, a transmembrane	Single signaling domain (CD3ζ)	Basic design, first proof-of-concept	Weak proliferation and activation, poor cytokine release, limited efficacy	[21,22,23,24,25,26,27,28]
Second	costimulatory molecule (CD28, CD134 etc.), CD3ζ, transmembrane.	Added costimulatory domain (CD28 or 4-1BB)	Improved T-cell proliferation, activation and survival; better persistence	Causes immunosuppressive tumor microenvironment (TME) due to single costimulatory molecule	[19,20,21,22]
Third	CD3ζ, costimulatory molecule (CD27, CD28, CD134 etc.); CD3ζ-CD28-CD134, CD3ζ-CD28-CD137 and CD3ζ TLR2-CD28	Dual costimulatory domains (CD28 + 4-1BB, or others)	Increase cytokine release, better antitumor activity	Rapid exhaustion of CAR-T cells; Toxicity due to over activation of costimulatory molecule, complex design	[25,26]
Fourth (TRUCKs)	costimulatory molecule (CD28, CD134 etc.), CD3ζ, transmembrane, Cytokine costimulatory molecule (IL-2, IL-5, IL-15)	Cytokine-secreting CARs (e.g., IL-12)	Activate innate response and enhance immune response, regulate TME modulation,	Increased side effect and high risk of offsite target	[27,28]
Fifth	costimulatory molecule (CD28, CD134 etc.), CD3ζ, transmembrane, TF binding motif and cytoplasmic receptor	Incorporates cytokine receptor signaling	Enhances immune response, better persistence and function	Still under early-stage research pregress, potential unknown risks	[30]

**Table 2 medsci-13-00190-t002:** Surface protein-targeted CAR-T-cell therapy for solid tumors in clinical development.

Target Antigens	Cancer	Treatment	Phase	Clinicaltrials.gov Identifier	Status
HER	R/R HER2-positive solid tumors	CAR-T cells (CCT303-406)	Phase I	NCT04511871	Active, not recruiting
	HER2-positive local advanced/metastatic solid tumors	Hypoxia-stimulated CAR HER2 T cells	Phase I, II	NCT05681650	Recruiting
	Brain and/or Leptomeningeal metastases	HER2-targeted CAR-T cells	Phase I	NCT03696030	Recruiting
MSLN	MSLN-positive ovarian cancer	TCR-like CAR-T cells	Phase I	NCT05963100	Recruiting
	MSLN-positive advanced malignant solid tumors	Anti-MSLN CAR-T cells	Phase I	NCT05783089	Not yet recruiting
	Pancreatic cancer	MSLN targeted CAR-T cells	Phase I	NCT05779917	Recruiting
	Advanced/Metastatic solid tumors	TAK-103	Phase I	NCT05164666	Active, not recruiting
	Advanced, locally advanced (inoperable) metastatic MSLN-expressing cancers	ZW171	Phase I	NCT06523803	Recruiting
GD2	High Risk and/or R/R Neuroblastoma	GD2 targeted CAR-T cells	Phase I, II	NCT03373097	Recruiting
	Lung Cancer	GD2 CAR-T cells + IL-15 + iCaspase9	Phase I	NCT05620342	Recruiting
	R/R Neuroblastoma	GD-2 CAR-T cell + iCaspase9	Phase I	NCT01822652	Active, not recruiting
MUC 1	MUC1 Positive Advanced Solid Tumors	Anti-CTLA-4/PD-1 expressing MUC1 CAR-T cells	Phase I/II	NCT03179007	Unknown status
	Advanced/Metastatic epithelial-derived solid tumors	P-MUC1C-ALLO1 CAR-T cells	Phase I	NCT05239143	Recruiting
MUC 16	R/R Epithelial Ovarian, Primary Peritoneal, or Fallopian Tube Cancer	Anti MUC16 CAR-T cell (27T51)	Phase I	NCT06469281	Recruiting
EGFR	Advanced adult NSCLC	CXCR5-modified EGFR CAR-T cells	Phase I	NCT05060796	Recruiting
	R/R Solid tumor in children and young adults	EGFR806-specific CAR-T cell	Phase I	NCT03618381	Recruiting

Abbreviations: EGFR = Epidermal growth factor receptor, GD2 = Diganlioside, HER = Human epidermal growth factor receptor, MUC = Mucin, MSLN = Mesothelin.

**Table 3 medsci-13-00190-t003:** Summaries in tabulated form of challenges and strategies associated CAR-T-cell therapy.

Category	Challenges	Strategies	Examples	Reference
Toxicities associated CAR-T Therapy	High toxicity rates limit first-line use	- Toxicity Management: Tocilizumab, corticosteroids, IL-6 blockade - Engineering CARs to reduce toxicity: Modify CAR-T-cell structures to reduce toxicity (e.g., cytokine secretion).	- Most common in 2nd-gen CD19 CAR-T therapy - CRS, ICANS, renal failure, MAS, splenomegaly - Higher-grade CRS in FDA-approved therapies (tisagenlecleucel, axicabtagene)	[62]
Antigen Escape	Loss of expression at antigens targets causes tumor resistance	- Targeting Multiple Antigens; construct Tandem CARs	- Develop dual-directed CAR-T cells (e.g., CD19/CD22, CD19/BCMA). Construct CARs with two different ScFvs targeting multiple antigens simultaneously.	[63]
On-Target/Off-Tumor Effects	Normal tissue damage due to shared antigen expression	- Careful antigen selection - Clinical trials with more tumor-specific targets (e.g., TAG72 CARs for ovarian cancer.	- Solid tumor antigens also present on healthy tissue - Risk of toxicity when selecting broadly expressed targets like B7H3, MUC1, MUC16, TAG72	[64,65]
Tumor Microenvironment (TME)	Immunosuppression and poor infiltration reduce CAR-T efficacy	- Combine with PD-1 inhibitors (e.g., NCT05659628, NCT03706326) - Gene editing, e.g., Siglec15 or cMet-CARs - CARs resistant to TME suppression	Presence of checkpoint proteins, suppressor cytokines, regulatory T cells - High cell density and low vascularity in solid tumors	[66,67]
Trafficking and Infiltration	CAR-T cells face barriers reaching and penetrating solid tumors	- Local delivery (intrapleural/intraventricular) - CARs with heparinase/FAP to degrade ECM - CARs expressing CXCR1/CXCR2 receptors	- Physical barriers (tumor stroma, ECM with HSPG) - Inefficient chemokine signaling	[68,69]

**Table 4 medsci-13-00190-t004:** Major ongoing clinical studies of CAR-T cell therapies and T-cell therapies.

Indication	Therapy	Phase	Sponsor	Clinical Trial
R/R Multiple Myeloma				
	Anti-GPRC5D CAR-T Cells	I/II	XuYan	NCT05749133
	BCMA-targeted CAR-T cells		Chongqing Precision Biotech Co., Ltd.	NCT04272151
	BCMA-targeted CAR-T Cells	I	University of California, San Francisco	NCT05577000
	CAR-T cells (Retreatment)	I	Second Affiliated Hospital of Xi’an Jiaotong University	NCT03672253
	BCMA-TGF-BETA CAR-T Cells	I	Medical College of Wisconsin	NCT05976555
	BCMA-targeted CAR-T Cells	I	Second Affiliated Hospital, School of Medicine, Zhejiang University	NCT04706936
	BCMA-targeted CAR-T Cells	I/II	Chongqing Precision Biotech Co., Ltd.	NCT04271644
	CS1-targeted CAR-T Cells	I	Zhejiang University	NCT04541368
Acute Myeloid Leukemia				
	CLL-1-, CD33- and/or CD123-specific CAR-gene-engineered T cells	I/II	Shenzhen Geno-Immune Medical Institute	NCT04010877
	Anti-CLL1 CAR-T cells	I	920th Hospital of Joint Logistics Support Force of People’s Liberation Army of China	NCT04923919
	CD123-targeted CAR-T Cells	I/II	Chongqing Precision Biotech Co., Ltd.	NCT04265963
Hematological Malignancies				
	GLPG CAR-T cells	III	Galapagos NV	NCT06652633
	CAR-T Autologous T-cell injection	NA	Hebei Senlang Biotechnology Inc., Ltd.	NCT05618041
	CAR-T cells	I	Mingzhi Zhang	NCT06647329
Acute Lymphoblastic Leukemia				
	Anti-CD7 CAR-T cells	I	National University Hospital, Singapore	NCT05043571
	CD19 and CD22 CAR sequential therapy	I/II	Chongqing Precision Biotech Co., Ltd.	NCT04265963
	CD5/CD7-specific CAR-T cells (BAH246)	I/II	Essen Biotech	NCT06420076
	CD19/CD22 CAR-T cells	NA	Beijing GoBroad Hospital	NCT06343090
R/R B-lineage	CAR-T cell	I	National University Hospital, Singapore	NCT05038696
R/R T-cell malignancies				
	CD4 CAR-T cells	I	iCell Gene Therapeutics	NCT04162340
	Humanized CD7 CAR-T cells	II	The First Affiliated Hospital of Soochow University	NCT05059912
	CD4-specific CAR-T Cells	I	iCell Gene Therapeutics	NCT04162340
R/R B-cell Malignancies				
	CD19+-Targeted CAR-T cells	I/II	Chongqing Precision Biotech Co., Ltd.	NCT04271410
	Cord-Blood-Derived CAR-T cells	I	Henan Cancer Hospital	NCT03881774
R/R Immune Thrombocytopenia				
	Anti-BCMA CAR-T cells	II	The First Affiliated Hospital of Soochow University	NCT05315778
Refractory SLE				
	CD19-BCMA CAR-T cells	I	RenJi Hospital	NCT05858684
Recurrent Glioblastoma				
	Tris-CAR-T cells	Phase	Beijing Tiantan Hospital	NCT05577091
Advanced Malignant Solid Tumors				
	TIL Therapy	I	Shanghai Juncell Therapeutics	NCT05417750
	TIL Therapy	I	Shanghai Juncell Therapeutics	NCT06488950
	TIL Therapy	I	Hervor Therapeutics	NCT06334783
	TIL Therapy	I	AgonOx, Inc.	NCT05902520
	TIL Therapy	I	Shanghai Juncell Therapeutics	NCT04967833
Pediatric High Risk Solid Tumors	TIL Therapy	I	Johns Hopkins All Children’s Hospital	NCT06047977
Advanced Melanoma				
	TIL Therapy	II	Shanghai Juncell Therapeutics	NCT06703398
	TIL Therapy	I/II	Shanghai Juncell Therapeutics	NCT06120712
	TIL Therapy	I	Shanghai Juncell Therapeutics	NCT05098184
Advanced Gynecologic Tumors				
	TIL Therapy	I	Shanghai Juncell Therapeutics	NCT05098171
	Membrane-Bound Cytokine-Modified TIL	I	Shanghai Juncell Therapeutics	NCT05468307
R/R Gynecologic Tumors	TIL Therapy	I	Shanghai 10th People’s Hospital	NCT04766320
NSCLC’s				
	TIL Therapy	Ib	Shanghai Juncell Therapeutics	NCT06473961
	TIL Therapy	II	University Hospital, Basel, Switzerland	NCT06455917

Abbreviations: BCMA = B-cell acute lymphoblastic leukemia, CAR = Chimeric antigen receptor, R/R = Relapsed or refractory, SLE = Systemic Lupus Erythematous, TIL = Tumor-infiltrating lymphocytes.

**Table 5 medsci-13-00190-t005:** Use of targeted TCR-engineered cells against various types of cancer.

Type of Cancer	Target Antigen	Clinical Trial Number	Reference
Melanoma	Melanoma Antigen Recognized by T cells-1 (MART-1)	NCT00910650	[104]
	New York Esophageal Squamous Cell Carcinoma-1 (NY-ESO-1)	NCT01967823	
	Melanoma-Associated Antigen (MAGE)-A3	NCT02111850	
Pancreatic Cancer	KRAS G12D	NCT03935893	[105]
Human Papilloma Virus (HPV)-associated epithelial cancer	E6 and E7 antigens of HPVs	NCT02858310	[106]
Non-small-cell lung cancer	MAGE-A10	NCT02592577	[107]
Ovarian cancer	Mesothelin	-	[108]
Sarcoma	NY-ESO-1	-	[109]
Renal Cell Carcinoma	TNF-related apoptosis-inducing ligand (TRAIL)	NCT00923390	[110]
Esophageal Cancer	MAGE-A4	-	[111]
Colorectal Cancer	Carcinoembryonic antigen (CEA)	-	[112]

**Table 6 medsci-13-00190-t006:** Comparative analysis of various types of ACT.

Characteristic	CAR-T-Cell Therapy	TIL Therapy	TCR Therapy
Target	Surface proteins (e.g., CD19, BCMA)	Neoantigens or tumor-specific antigens	Intracellular peptides bound to MHC
Specificity	Low	High	High
MHC Dependency	No	Yes	Yes
Tumor Types Treated	Hematological malignancies (e.g., B-cell leukemia, lymphoma)	Solid tumor, including melanoma, ovarian, cervical, etc.	Solid tumors and hematologic malignancies
Time to Prepare	Several weeks (complex engineering)	Several weeks (tumor extraction and expansion)	Several weeks (engineering and selection of TCRs)
Toxicity	CRS, neurotoxicity	IL-2-related toxicity. Hypotension, nausea	Off-target effects, CRS

## Data Availability

No new data were created or analyzed in this study.

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
