# Peer review of "Advances in Adoptive Cell Therapies in Cancer: From Mechanistic Breakthroughs to Clinical Frontiers and Overcoming Barriers"

_medsci, 2025, doi:10.3390/medsci13030190_

Round 1
Reviewer 1 Report
Comments and Suggestions for Authors
In this article, Rabbani et al. have discussed an overview of adoptive cell therapies (ACTs) in cancer treatment, primarily focusing on chimeric antigen receptor (CAR)-T cells, tumor-infiltrating lymphocytes (TILs) and T-cell receptor (TCR)-engineered T cells. The authors have also highlighted challenges such as antigen escape, toxicity, and the immunosuppressive tumor microenvironment, presenting innovative strategies to overcome these hurdles.
Although, the manuscript is timely, well-constructed, extensively researched and informative, in the current form, the article can benefit from certain minor elements.
Recommendations:
- Throughout the review the authors have denoted CAR T in multiple ways. CART, CAR T and CAR-T. Authors need to be consistent with nomenclature.
- The authors should elaborate about Neoantigens recognition by T cells.
- There should ideally be a dedicated section on T cell exhaustion and its discussion in regard to some of its markers (like PD1, TIM3 and LAG3)
- Section 1 and 2 can be combined into one introduction section.
- The authors mention in the future perspective section about AI/ML. It would be nice if a small section is added to this review. It will add to the uniqueness of the manuscript and make it more current and relevant.
Author Response
Reviewers’ Comments
Reviewer 1
- Throughout the review the authors have denoted CAR T in multiple ways. CART, CAR T and CAR-T. Authors need to be consistent with nomenclature.
Response:
We thank the reviewer for pointing out the inconsistency in the nomenclature. We have carefully revised the manuscript and standardized the terminology to “CAR-T” throughout the text to ensure uniformity and clarity.
- The authors should elaborate about Neoantigens recognition by T cells.
Response:
We thank the reviewer for highlighting the need to elaborate on neoantigen recognition by T cells. In the revised manuscript, we have incorporated a concise yet comprehensive section within the “Fundamentals of CAR- T cell therapy” part. This addition strengthens the mechanistic foundation of the review and aligns with the translational focus of the manuscript.
Modifications done are highlighted on page no. 3
3) There should ideally be a dedicated section on T cell exhaustion and its discussion in regard to some of its markers (like PD1, TIM3 and LAG3).
Response:
We thank the reviewer for this valuable suggestion. In the revised manuscript, we have added a dedicated section on T-cell exhaustion, highlighting its biological basis, clinical impact on adoptive cell therapy, and key exhaustion markers such as PD-1, TIM-3, and LAG-3. This addition strengthens the mechanistic depth of the review and provides a clearer link between T-cell biology and therapeutic performance.
To address this, we have added a subsection 2.3.2 titled as ‘T-Cell Exhaustion’
Modifications done are highlighted in text on page no 11.
4) Section 1 and 2 can be combined into one introduction section.
Response:
We appreciate the reviewer’s suggestion. Sections 1 and 2 have been merged into a single comprehensive Introduction section, ensuring a logical flow without redundancy. All original content and reference citations have been retained, and transitions have been refined for coherence.
Modifications done are highlighted in text on page no.2, 3
5) The authors mention in the future perspective section about AI/ML. It would be nice if a small section is added to this review. It will add to the uniqueness of the manuscript and make it more current and relevant.
Response:
We appreciate the reviewer’s valuable suggestion. In response, we have incorporated a dedicated subsection titled “Integration of Artificial Intelligence and Machine Learning in CAR T-Cell Development” and “Integration of Artificial Intelligence and Machine Learning in TCR-Engineered T Cells”. This addition highlights how AI/ML approaches are increasingly being applied to optimize CAR/TCR design, enhance manufacturing quality control, and predict therapy-associated toxicities, thereby underscoring their potential to accelerate and personalize adoptive cell therapy development.
To address this, we have revised the manuscript.
Modifications done are highlighted in the text on page no.14, 21, 24.
Reviewer 2 Report
Comments and Suggestions for Authors
The authors present an over-arching review of Adoptive Cell Therapies for cancer treatment. However, the review suffers from a lack of focus and details. The authors need to work on and address the following:
- The manuscript has multiple formatting, spelling, and grammatical errors that need to be addressed:
- Multiple words are oddly hyphenated, e.g. line 43 – “ex-hibit”. Please find and correct all of these.
- The figures are usually extremely low resolution and are very inconsistently formatted and drawn. Some figures are clearly made using Biorender (please cite) while some are made in Powerpoint. The figures need to be consistent and should stylistically look the same.
- The grammar and style of the different sections vary significantly. Ideally, the lead author should take care to go through the manuscript and ensure that the writing style and grammar are consistent.
- Please comb through the manuscript to make sure that everything is grammatically sound and spelled correctly.
- 100 – what do you mean by cancer-testis?
- 125-129 – Split into multiple sentences for clarity. “In this technology, T lymphocytes cells are engineered with desired genes after isolating them from circulation to recognize and combat cancer cells without engaging MHC. The modified cells are proliferated ex vivo, following which they are infused back to the patient to drive an immune response against tumors and monitored for the persistence of CAR T cells and for the disease response as indicated in Fig 1.”
- The authors keep mentioning the limitations of CAR T cells in solid tumors. Mentioning some aspects of why this is a problem will be beneficial.
- 228 – why is tumor lysis considered a toxic effect?
- 2.4 – Multiple Myeloma. There is no discussion of CAR T-cells in this context, hence the paragraph can be removed.
- 297 – Can you elaborate why blinatumomab is being discussed in the context of CAR T-cells?
- Section 4 – Please put quantifications for the outcomes here. In general, the review is lacking in details of outcomes for the different treatments. Please include outcomes such as survival, relapse etc.
- Future Directions and Conclusions – The inclusion of the use of AI and ML for ACT development seems odd, since this has not been discussed in the main body of the paper at all. Can the authors include a section of this in the manuscript?
The authors need to ensure that the quality of English is polished more. There are several grammatical, spelling, and formatting errors that need to be fixed. Please refer to the previous comments for more details on these.
Author Response
Reviewers’ Comments
Reviewer 2
- The manuscript has multiple formatting, spelling, and grammatical errors that need to be addressed:
- Multiple words are oddly hyphenated, e.g. line 43 – “ex-hibit”. Please find and correct all of these.
- The figures are usually extremely low resolution and are very inconsistently formatted and drawn. Some figures are clearly made using Biorender (please cite) while some are made in Powerpoint. The figures need to be consistent and should stylistically look the same.
- The grammar and style of the different sections vary significantly. Ideally, the lead author should take care to go through the manuscript and ensure that the writing style and grammar are consistent.
- Please comb through the manuscript to make sure that everything is grammatically sound and spelled correctly.
Response:
We thank the reviewer for this valuable feedback. We have thoroughly revised the manuscript to ensure uniformity in formatting, spelling, and grammar. All instances of inadvertent hyphenation (e.g., “ex-hibit”) have been corrected. The figures have been added with high resolution in consistent style and formatting; those created using BioRender are now appropriately cited. We have also standardized the writing style across sections to maintain consistency in tone and grammar. A comprehensive language edit was performed to ensure grammatical accuracy and clarity throughout the manuscript.
- 100 – what do you mean by cancer-testis?
Response:
We appreciate the reviewer’s request for clarification. Cancer–testis antigens (CTAs) are a distinct group of tumor-associated antigens whose expression is normally restricted to germ cells of the testis (and occasionally ovary) but aberrantly re-expressed in various malignancies. Importantly, because germ cells lack MHC class I expression, CTAs are not subject to central immune tolerance, making them highly immunogenic and attractive targets for T-cell–based immunotherapies, including CAR-T and TCR-T strategies.
3) 125-129 – Split into multiple sentences for clarity. “In this technology, T lymphocytes cells are engineered with desired genes after isolating them from circulation to recognize and combat cancer cells without engaging MHC. The modified cells are proliferated ex vivo, following which they are infused back to the patient to drive an immune response against tumors and monitored for the persistence of CAR T cells and for the disease response as indicated in Fig 1.”
Response:
We thank the reviewer for this valuable suggestion. The sentence in question (Lines 125–129) has been revised for improved clarity and readability by splitting it into multiple sentences.
Modifications done are highlighted in text on page no 3.
4) The authors keep mentioning the limitations of CAR T cells in solid tumors. Mentioning some aspects of why this is a problem will be beneficial.
Response:
We appreciate the reviewer’s suggestion. We have now elaborated on the specific challenges that limit the efficacy of CAR T cells in solid tumors, adding mechanistic insights to clarify the problem.
Modifications done are highlighted in the text on page no.13, 14
5) 228 – why is tumor lysis considered a toxic effect?
Response:
We thank the reviewer for pointing out the need for clarification. Tumor lysis is considered a toxic effect because the rapid destruction of cancer cells—often seen after potent immunotherapies like CAR T cells—releases intracellular contents such as potassium, phosphate, and uric acid into the bloodstream. This can precipitate tumor lysis syndrome (TLS), characterized by acute kidney injury, electrolyte imbalances, arrhythmias, seizures, and, in severe cases, death. While tumor lysis reflects effective anti-tumor activity, the uncontrolled biochemical overload poses a serious, potentially life-threatening toxicity.
6) 2.4 – Multiple Myeloma. There is no discussion of CAR T-cells in this context; hence the paragraph can be removed.
Response:
We thank the reviewer for this observation. We have revised the paragraph to directly integrate the relevance of CAR T-cell therapy in the context of multiple myeloma. This update aligns the section with the manuscript’s focus and ensures a direct discussion of CAR T-cell applications.
Modifications done are highlighted in the text on page no.8
7) 297 – Can you elaborate why blinatumomab is being discussed in the context of CAR T-cells?
Response:
We thank the reviewer for this comment. We have now expanded the section on antigen escape to include a brief discussion of blinatumomab as a complementary antigen-targeting immunotherapy. This addition highlights its mechanism of action, key differences from CAR T-cell therapy, and its relevance in the broader context of strategies to overcome antigen escape.
Modifications done are highlighted in the text on page no.11
8) Section 4 – Please put quantifications for the outcomes here. In general, the review is lacking in details of outcomes for the different treatments. Please include outcomes such as survival, relapse etc.
Response:
We appreciate the reviewer’s observation regarding the need for quantitative outcome data in Section 4 now as section 3. In the revised manuscript, we have included data from pivotal clinical trials, such as overall response rates (ORR), complete remission (CR) rates, and median progression-free survival (PFS), overall survival (OS), and relapse frequencies where available. This quantitative detail provides a more robust comparison between therapeutic approaches and enhances the clinical relevance of the review.
Modifications done are highlighted in the text on page no.14, 15, 16
9) Future Directions and Conclusions – The inclusion of the use of AI and ML for ACT development seems odd, since this has not been discussed in the main body of the paper at all. Can the authors include a section of this in the manuscript?
Response:
We thank the reviewer for this important observation. To address this, we have now added a dedicated separate subsection titled “Integration of Artificial Intelligence and Machine Learning in TCR-Engineered T Cells and CAR-T cell” within the main body of the manuscript. This section discusses the emerging role of AI/ML in epitope prediction, TCR design optimization, and personalized neoantigen identification. By embedding this discussion within the relevant therapy context, we ensure that the mention of AI/ML in the future perspectives is grounded in the core review narrative, enhancing coherence and relevance.
Modifications done are highlighted in the text on page no.14,21
Reviewer 3 Report
Comments and Suggestions for Authors
Suggestions for improving your submission:
- You could delve deeper into the economic and logistical challenges of ACTs, especially in low-resource settings.
- While the clinical outcomes are discussed, the patient experience and quality-of-life improvements could be explored further.
Minor:
Please check the text for formatting consistency and also add a glossary of terms/acronyms.
Author Response
Reviewers’ Comments
Reviewer 3
- You could delve deeper into the economic and logistical challenges of ACTs, especially in low-resource settings.
Response:
We appreciate the reviewer’s suggestion to address the economic and logistical challenges of ACTs, particularly in low-resource settings. We have now incorporated concise discussions of these aspects for CAR-T cell therapy and TCR-engineered T-cell therapy, emphasizing cost drivers, infrastructure requirements, and potential strategies for scalability and accessibility. These additions highlight the real-world barriers to implementation and the innovations being pursued to overcome them.
Modifications done are highlighted on page no. 14, 20, and 21 in the manuscript
2) While the clinical outcomes are discussed, the patient experience and quality-of-life improvements could be explored further.
Response:
We thank the reviewer for this valuable suggestion. We have now included a concise discussion of patient experience and quality-of-life (QoL) considerations for both CAR T-cell and TCR-engineered T-cell therapies, covering aspects such as symptom relief, functional recovery, treatment-related toxicities, and long-term well-being. These additions provide a more patient-centered future perspective, complementing the clinical outcome data already presented.
To address this, we have carefully gone through the manuscript to ensure best suitable column heading and formatting.
Modifications done are highlighted on page no.24
3) Please check the text for formatting consistency and also add a glossary of terms/acronyms.
Response:
We have thoroughly reviewed the manuscript for formatting consistency, ensuring uniform heading styles, font usage, citation format, and terminology across sections. In addition, we have standardized the use of acronyms and corrected any inconsistencies in figure/table captions.
Round 2
Reviewer 2 Report
Comments and Suggestions for Authors
I would like to thank the authors for incorporating my suggestions and providing a cleaner and more consistent version of the manuscript. There are a few minor edits still to be made.
- Lines 302 - 307: This paragraph should not be within quotes.
- Line 594 - 617: Please format this section to be consistent with the rest of the article, including using bullet points where applicable.
- Please ensure that all Biorender generated images are properly licensed and cited as detailed in this article: https://help.biorender.com/hc/en-gb/articles/21282341238045-How-to-download-a-publication-license-for-open-access-resources
Author Response
Reviewer comment
Reviewer comment
- Lines 302 - 307: This paragraph should not be within quotes.
Response:
We thank the reviewer for pointing this out. The paragraph has now been revised and presented without quotation marks to ensure consistency with the manuscript style. Appropriate citations have been retained to credit the original sources.
Modifications done are highlighted on page no.11
- Line 594 - 617: Please format this section to be consistent with the rest of the article, including using bullet points where applicable.
Response:
Thank you for this valuable suggestion. We have revised the section (lines 594–617) to ensure formatting consistency with the rest of the manuscript. Specifically, the content has been provided proper subsections for better readability and uniformity with the formatting style used throughout the article as per the direction of reviewer.
Modifications done are highlighted on page no.19, 20, 21
- Please ensure that all Biorender generated images are properly licensed and cited as detailed in this article https://help.biorender.com/hc/en-gb/articles/21282341238045-How-to-download-a-publication-license-for-open-access-resources.
Response:
We thank the reviewer for this important observation. We have included the recommended citation “Created with BioRender.com” in the figure legends to ensure compliance with licensing and citation requirements as per the direction of reviewer. We have already shared the licenses of all the figures with the journal.
Modifications done are highlighted in each figure legends on page no. 4,5,8,13,15.